# Infectious complications related to radiofrequency ablation of liver tumors: The role of antibiotics

Ryo Nakagomi[1], Ryosuke Tateishi[1]*, Shintaro Mikami[1], Taijiro Wake[1], Mizuki Nishibatake Kinoshita[1], Takuma Nakatsuka[1], Tatsuya Minami[1], Masaya Sato[1], Koji Uchino[1], Kenichiro Enooku[1], Hayato Nakagawa[1], Yoshinari Asaoka[2], Shuichiro Shiina[3], Kazuhiko Koike[1]

1 Department of Gastroenterology, Graduate School of Medicine, The University of Tokyo, Tokyo, Japan,
2 Department of Medicine, Teikyo University School of Medicine, Tokyo, Japan, 3 Department of Gastroenterology, Juntendo University Hospital, Tokyo, Japan

* tateishi-tky@umin.ac.jp

**Data Availability Statement:** All relevant data are within the paper and its Supporting information files.

## Abstract

### Background and aim

Prophylactic administration of antibiotics within 24 hours of surgery is recommended to reduce the risk of infection. We conducted a prospective study to compare the efficacy of single administration of antibiotics with a historical control of continuous administration of antibiotics for radiofrequency ablation (RFA) of malignant liver tumors.

### Methods

Between February 1, 1999 and November 30, 2010, a total of 6,763 RFA treatments were performed in 2,355 patients, using a protocol with continuous administration of prophylactic antibiotics. On December 1, 2010, we began using a revised protocol with a single administration of prophylactic antibiotics, while continuing to use the old continuous administration protocol for patients who declined the new protocol. Interim analysis was performed to assess the safety of the single administration protocol. Thereafter, from April 1, 2012, all patients were treated using the new protocol. Risk factors for infectious complications of RFA were assessed using logistic regression.

### Results

From December 2010 to March 2012, 766 RFA treatments were performed in 663 patients using the new antibiotic protocol. Infectious complications were observed following 4 of these treatments (0.52%). As the upper limit of the confidence interval (CI) resulting from a one-sided binomial test was exactly the prespecified limit of 1.0%, from April 2012 onwards, we treated all patients using the new protocol with single administration of prophylactic antibiotics. A total of 3,547 RFA treatments were performed using the single administration protocol. Univariable logistic regression indicated that prior transcatheter arterial chemoembolization (TACE) and maximal tumor diameter were significant risk factors for infectious

**Funding:** This work was supported by Health, Labour and Welfare Policy Research Grants from the Ministry of Health, Labour and Welfare of Japan (Policy Research for Hepatitis Measures [H30-Kansei-Shitei-003]).

**Competing interests:** The authors have declared that no competing interests exist.

**Abbreviations:** ABPC/SBT, ampicillin/sulbactam; ALT, alanine aminotransferase; AST, aspartate aminotransferase; CI, confidence interval; CT, computed tomography; DM, diabetes mellitus; ERCP, endoscopic retrograde cholangiopancreatography; FMOX, flomoxef sodium; HCC, hepatocellular carcinoma; HCV, hepatitis C virus; INR, international normalized ratio; IQR, interquartile range; MRI, magnetic resonance imaging; RFA, radiofrequency ablation; SD, standard deviation; TACE, transcatheter arterial chemoembolization.

complications (P = 0.04 and P < 0.001, respectively). Multivariable analysis indicated that the adjusted hazard ratio of single vs. continuous administration of antibiotics was 1.20 (95% CI: 0.53–2.75; P = 0.66).

## Conclusions

The rate of infectious complications related to RFA was acceptably low. Single administration of prophylactic antibiotics did not significantly increase the rate of infectious complications related to RFA, compared with a more intensive antibiotic protocol.

## Introduction

Percutaneous tumor ablation of hepatocellular carcinoma (HCC) is an important nonsurgical option with a high rate of success that can preserve liver function [1–4]. Radiofrequency ablation (RFA) is among the most commonly employed methods of percutaneous tumor ablation, as an efficient technique with a high rate of local success [5–9].

Because RFA is a commonly used thermal ablative procedure, a great deal of data is available regarding complications related to RFA [10–13]. Of these, infectious complications such as liver abscess have been reported to be the third most frequent major complication, with only tumor seeding and intraperitoneal bleeding occurring more frequently. According to a systematic review, liver abscess occurs in 0.32% of patients who undergo RFA [13]. Choi et al. reported a higher rate of liver abscess: 1.7% in their early experience of 603 patients [14]. They identified preexisting biliary abnormality prone to ascending biliary infection, tumor with retention of iodized oil from previous transcatheter arterial chemoembolization (TACE), and treatment with an internally cooled electrode as risk factors for abscess formation after RFA.

Previous studies have focused mainly on the assessment of infectious complications and their countermeasures. However, there has been little discussion regarding prophylactic procedures, including antibiotic use. In the aforementioned report by Choi et al., antibiotics were not administered prior to ablation [14]. The Centers for Disease Control and Prevention propose the administration of prophylactic antibiotics 1 hour prior to incision, followed by supplementary injections every 3–4 hours throughout the operation, and they do not recommend the extended use of antibiotics beyond 24 hours [15, 16]. Shorter duration of antibiotic use may reduce the risk of emergence of resistant strains.

Following the introduction of ethanol injection in 1985, we adopted a protocol of continuous administration of antibiotics for 24 hours as prophylaxis to prevent infection following percutaneous ablation of liver tumors [7, 17]. However, it is not clear whether continuous administration of antibiotics reduces the risk of infectious complications related to RFA, compared with single administration. We conducted a trial of prophylactic single administration of antibiotics with a historical control of continuous administration of antibiotics, to compare the frequency and risk factors of infectious complications following RFA.

## Patients and methods

### Study protocol

Before the start of the present study, between February 1, 1999 and November 30, 2010, a total of 6,763 RFA treatments were performed in 2,355 patients with liver malignancy in the Department of Gastroenterology, the University of Tokyo Hospital. In those days, we administered continuous prophylactic antibiotics until fever reduced following RFA. Infectious complications were diagnosed following 15 of these treatments (0.22%). Since this incidence was too

low to conduct a randomized controlled trial, we decided to conduct a prospective study with a historical control. We began using a revised protocol with a single administration of prophylactic antibiotics on December 1, 2010. Patients who gave written informed consent to the new protocol were administered antibiotics once prior to ablation, while other patients were treated with the old protocol of continuous administration. Interim analysis was performed in March 2012 to assess the safety of the new protocol. Thereafter, all patients were treated using the new protocol.

This study was conducted according to the ethical guidelines for epidemiological research designed by the Japanese Ministry of Education, Culture, Sports, Science and Technology and the Ministry of Health, Labour and Welfare. The study protocol was approved by the University of Tokyo Medical Research Center Ethics Committee and registered with the University Hospital Medical Information Network (UMIN) Clinical Trial Registry (UMIN000004974).

## Prophylactic antibacterial drug administration

Prior to this prospective study, we used flomoxef sodium or cefmetazole in a continuous administration regimen. For the single administration regimen, we chose ampicillin/sulbactam. When a patient displayed symptoms consistent with infectious complications, or an increase in fever over a number of days, we resumed antibiotic administration.

## Radiofrequency ablation

Inclusion criteria for RFA treatment in the authors' institution were as follows: tumor size not larger than 3 cm in diameter, number of tumors not more than three, absence of vascular invasion and extrahepatic metastasis, serum total bilirubin concentration not higher than 3 mg/dL, platelet count no less than $50 \times 10^3/mm^3$, prothrombin activity no less than 50% (approximately 1.5 INR), and absence of uncontrollable ascites and hepatic encephalopathy. We also excluded patients who had a history of bilioenteric anastomosis or sphincterotomy that are considered as high risk for hepatic abscess formation. A detailed protocol for RFA is described elsewhere [7]. A session was defined as a single intervention episode that consisted of one or more ablations performed on one or more tumors, and a treatment was defined as the completed effort to ablate one or more tumors that consisted of one or more sessions. RFA was performed on an inpatient basis.

## Diagnosis of infectious complications after RFA

Typical infectious complications that we observed during the trial and post-trial phases were liver abscess, cholecystitis, and cholangitis. Diagnosis of liver abscess was made by clinical examination, laboratory findings, and radiographic imaging, followed by aspiration under ultrasonography and culture of the abscess material. Cholecystitis was diagnosed when the gallbladder was swollen, with wall thickening, accompanied by hypochondriac tenderness. Cholecystitis was confirmed by percutaneous aspiration. Cholangitis was diagnosed based on enhanced CT (computed tomography) or MRI (magnetic resonance imaging) and laboratory findings, then confirmed by endoscopic retrograde cholangiopancreatography (ERCP).

## Statistical analysis

We analyzed the incidence of infectious complications on a per treatment basis, where a treatment was defined as the completed effort to ablate one or more tumors, consisting of one or more sessions according to international guidelines for tumor ablation [18].

We set the incidence limit to 1% for the safety assessment of the new protocol. We calculated that a sample size of 751 would be necessary to prove an incidence of no more than 1%, under the assumption that the true incidence would be 0.3%, using a one-sided binomial test with a type I error level of 5% and 80% statistical power. We estimated that some patients would be lost to follow-up and therefore set the number of treatments required for the safety assessment to 760.

Data are expressed as medians and 25th to 75th percentiles or frequencies and percentages, unless otherwise indicated. Confidence intervals (CI) for the rate of infectious complications were calculated based on a binomial distribution. A one-sided binominal test was used for the safety assessment of a single administration of prophylactic antibiotics. The variables assessed in the analysis of risk factors for infectious complications were as follows: age, sex, primary site of tumor, maximal tumor diameter, number of tumors, serum albumin concentration, serum total bilirubin concentration, aspartate aminotransferase concentration, alanine transaminase concentration, platelet count, presence of diabetes mellitus, and mode of antibiotic administration (single vs. continuous). Univariable and multivariable logistic regression analyses were performed. All statistical analyses were performed using R software (version 3.2.0; R Foundation for Statistical Computing, Vienna, Austria).

## Results

### Patient characteristics

Between December 2010 and March 2012, 663 patients gave informed consent to participate in the prospective study, and 766 RFA treatments were performed using the new protocol of single administration of antibiotics. During the same period, 279 patients did not give informed consent and were therefore treated with the old protocol of continuous administration of antibiotics, with 348 treatments. As a result, the continuous administration group comprised a total of 7,111 treatments. From April 2012, we treated all patients with a single administration of prophylactic antibiotics. The single administration group therefore comprised a total of 3,547 treatments (Figs 1 and 2).

Baseline characteristics of the patients are shown in Table 1. Patients were significantly older in the continuous administration group, and the proportion of patients who underwent TACE was significantly higher in the continuous administration group. The maximal tumor size was smaller in the single administration group.

### Infectious complications

In the continuous administration group, infectious complications were observed following 15 treatments (0.21%; 95% CI, 0.12–0.35%): liver abscess in 14 cases and cholecystitis in 1 (Table 2). The median (range) maximal tumor diameter was 3.2 (0.9–8.3) cm. The median (range) interval between the RFA procedure and diagnosis of infectious complications was 36 (1–385) days. Sixteen bacterial strains were identified in 11 of the 15 patients with infectious complications: *Enterococcus* in 8, *Pseudomonas* in 2, *Enterobacter* in 2, *Bacteroides* in 2, *Klebsiella* in 1, and *Escherichia coli* in 1. One patient recovered after antibiotic administration was resumed, without further intervention, while 14 patients underwent drainage (Fig 3); 2 of these patients died from infectious complications.

In the trial phase, infectious complications were observed following 4 treatments (0.52%; 95% CI, 0.14–1.3%): liver abscess in 3 cases and cholecystitis in 1. None of these patients died of infectious complications associated with RFA. The upper limit of the CI resulting from a one-sided binomial test exactly matched the preset incidence limit of 1.0%. We therefore adopted the single administration protocol for all patients who underwent RFA thereafter.

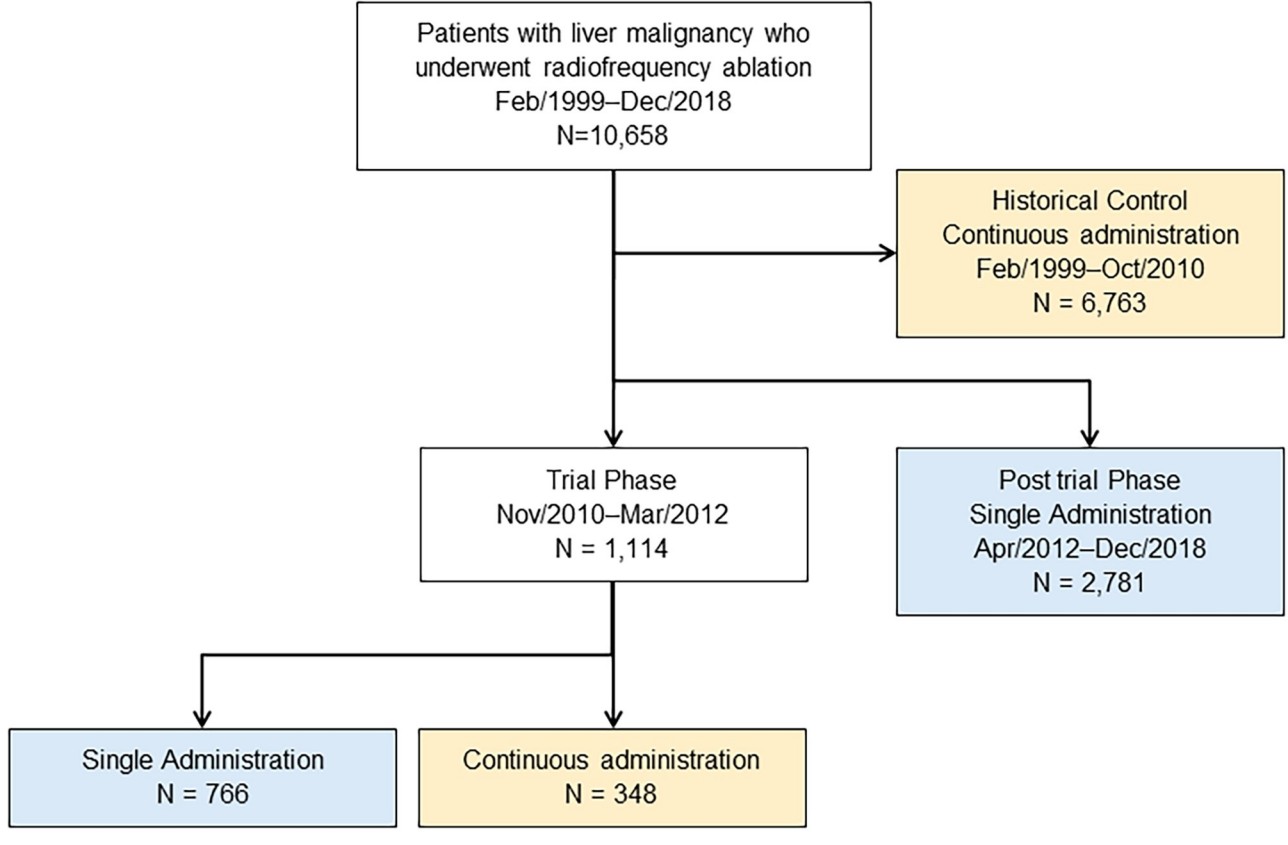

**Fig 1. Participant flow diagram.**

The post-trial cohort consisted of 2,781 treatments, following which infectious complications occurred in 5 cases (0.18%; 95% CI, 0.058–0.42%): liver abscess in all 5 cases. When RFA treatments using the single administration protocol in the trial and post-trial phases were combined, the total incidence of infectious complications was 0.25% (9 cases: 95% CI, 0.12–0.48%). The median (range) maximal tumor diameter was 2.6 (1.5–4.0) cm. The median interval between the RFA procedure and diagnosis of infectious complications was 14 (1–571) days. Eight bacterial strains were identified in 6 of the 9 patients with infectious complications: *Enterococcus* in 2, *Enterobacter* in 1, *Streptococcus* in 1, *Serratia* in 1, and *Escherichia coli* in 3. One patient recovered after antibiotic administration was resumed, without further intervention, while 8 patients underwent drainage; 1 of these patients died from infectious complications.

### Risk factors for infectious complications

We examined the risk factors for infectious complications using logistic analysis (Table 3). As a total, 3490 individual patients were included in the current study; 1951 were solely in the continuous administration group, 856 were solely in the single administration group, and 683 were in both groups. Univariable analysis indicated that prior TACE and maximal tumor diameter were significant factors ($P = 0.04$ and $< 0.001$, respectively). The adjusted hazard ratio of single vs. continuous administration of antibiotics was 1.20 (95% CI: 0.53–2.75; $P = 0.66$). When multivariable analysis was performed, only maximal tumor diameter

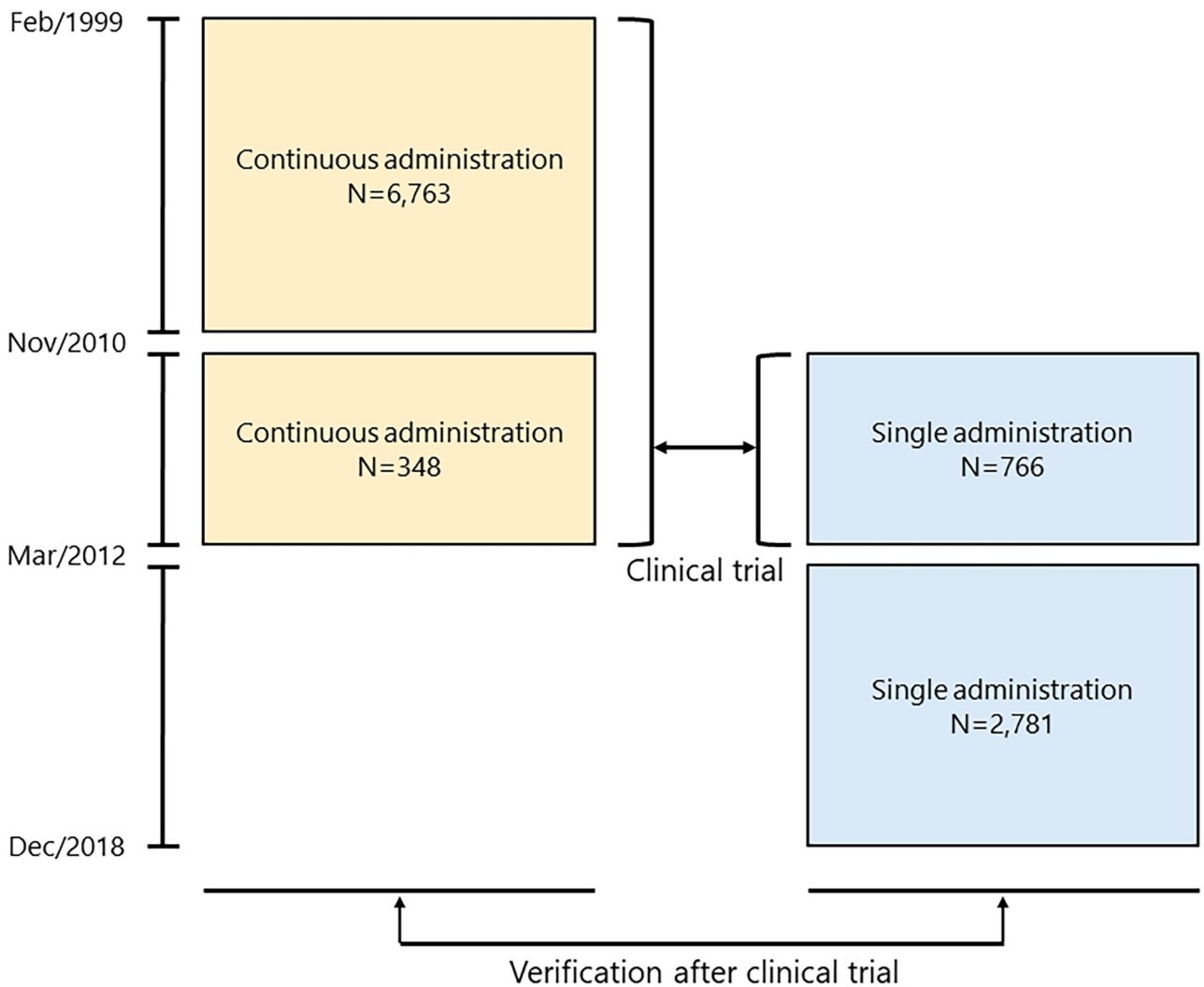

**Fig 2. Study outline.**

remained a significant factor ($P < 0.001$). An ad-hoc analysis with a dataset excluding overlap cases showed similar results (Table 4).

## Discussion

During the entire study period from February 1999 to December 2018, infectious complications were diagnosed following 25 RFA treatments (0.23%). Incidence of infectious complications was similar between treatments involving continuous and single administration of antibiotics. From a cost-benefit standpoint, continuous antibiotic use may therefore not be necessary. Furthermore, considering that nearly 50% of infectious complications occurred more than 2 weeks after RFA, antibiotic use may not be prophylactic at all, although this study cannot provide evidence for nonuse of prophylactic antibiotics.

Although infectious complications were generally controlled well with percutaneous drainage, 3 fatal cases occurred during the entire study period. All 3 of these patients underwent transhepatic abscess drainage. One patient died from multiple organ failure due to worsening

**Table 1. Baseline characteristics.**

| Characteristic | Continuous | Single | | P-value* |
|---|---|---|---|---|
| | | Trial phase | Post-trial phase | |
| Number of treatments | 7,111 | 766 | 2,781 | |
| Age (years), Mean ± SD | 70.0 ± 9.1 | 72.2 ± 9.4 | 74.0 ± 9.5 | <0.01 |
| Sex, n (%) | | | | 0.60 |
| Male | 4,727 (66.5) | 502 (65.5) | 1,837 (66.1) | |
| Female | 2,384 (33.5) | 264 (34.5) | 944 (33.9) | |
| Tumor type, n[†] | | | | |
| Hepatocellular carcinoma | 6,462 | 692 | 2,638 | |
| Intrahepatic cholangiocarcinoma | 15 | 5 | 14 | |
| Metastatic liver cancer | 636 | 73 | 135 | |
| Prior TACE, n (%) | 1,325 (18.6) | 27 (3.5) | 160 (5.8) | <0.01 |
| Maximal tumor diameter (cm)[‡] | 1.9 (0.4–13.5) | 1.6 (0.5–4.5) | 1.5 (0.3–6.2) | <0.01 |
| Number of nodules, n[‡] | 2 (1–22) | 1 (1–10) | 1 (1–13) | <0.01 |
| Serum albumin (g/dL)[‡] | 3.6 (1.2–5.1) | 3.6 (2.2–5.0) | 3.7 (1.9–5.1) | 0.16 |
| Total bilirubin (mg/dL)[‡] | 0.8 (0.2–5.9) | 0.9 (0.2–6.3) | 0.9 (0.2–4.3) | 0.50 |
| AST (IU/L)[‡] | 49 (6–429) | 42 (12–312) | 37 (4–296) | <0.01 |
| ALT (IU/L)[‡] | 39 (3–489) | 32 (4–315) | 27 (4–224) | <0.01 |
| Platelet count (× $10^4$/mm$^3$)[‡] | 10.9 (1.4–66.9) | 11.5 (2.1–44.0) | 11.7 (1.6–46.8) | 0.56 |
| FIB-4 Index[‡] | 1.63 (1.01–2.49) | 1.62 (1.11–2.40) | 1.71 (1.16–2.57) | 0.60 |
| APRI score[‡] | 1.04 (0.54–1.85) | 0.88 (0.46–1.56) | 0.75 (0.42–1.32) | <0.01 |
| DM, n (%) | 1,564 (22.0) | 208 (27.2) | 781 (28.1) | <0.01 |

* Comparisons were made between the continuous and single administration protocols using the t-test.

† Overlapping cases are included.

‡ Expressed as median (IQR).

Abbreviations: ALT, alanine transaminase; APRI, aspartate aminotransferase to platelet ratio index; AST, aspartate aminotransferase; DM, diabetes mellitus; FIB-4, Fibrosis-4; IU, international unit; IQR, interquartile range; SD, standard deviation; TACE, transcatheter arterial chemoembolization.

sepsis, and the other two patients died from acute respiratory distress syndrome caused by sepsis. The intervals between diagnosis of liver abscess and death in the three patients were 60, 77, and 89 days. It must be understood that, in some cases, infection cannot be controlled even with abscess drainage.

Consistent with previous reports, maximal tumor diameter and prior TACE were identified as risk factors for infectious complications [14, 19]. It is reasonable that maximal tumor diameter should be a risk factor, as a larger necrotic area is more susceptible to infection. TACE also results in a larger ablative zone with a decreased heat sink effect due to occlusion of intrahepatic blood flow [20], which may also potentially lead to infectious complications. In addition, ischemia caused by arterial embolization affects the intrahepatic bile ducts [21], which may increase the risk of infection from the intestine via the bile ducts.

Continuous administration of antibiotics may increase the risk of developing resistant strains. In fact, the strains detected in culture differed between the two treatment groups. *Enterococcus*, which possesses an intrinsic resistance to broad-spectrum antibiotics, was detected more commonly in the continuous administration group, and *Pseudomonas* was detected only in this group.

Although we included more than 10,000 RFA treatments in this study, the incidence of infectious complications was not high enough to provide sufficient statistical power. This may be because this study was performed in a high-volume center, where the incidence of

**Table 2. Patients with infectious complications after radiofrequency ablation (N = 24).**

| Patient No. | Group | Complication | Age | Sex | Tumor type | Etiology | DM | Maximal tumor diameter (cm) | Number of nodules (n) | Interval between RFA and diagnosis of complication (days) | Treatment for complication | Bile culture | Blood culture | Outcome |
|---|---|---|---|---|---|---|---|---|---|---|---|---|---|---|
| 1 | C | LA | 68 | M | HCC | HCV | + | 3.2 | 4 | 62 | Drainage | P. aeruginosa | Not taken | Recovered |
| 2 | C | LA | 61 | F | HCC | HBV | - | 7.6 | 2 | 36 | Drainage | E. faecium | No growth | Recovered |
| 3 | C | LA | 74 | M | HCC | HCV | + | 4.9 | 12 | 369 | Drainage | Not taken | A. sobria | Recovered |
| 4 | C | LA | 67 | M | HCC | HBV | + | 2.2 | 4 | 5 | Drainage | E. faecium | E. faecium | Recovered |
| 5 | C | LA | 83 | M | HCC | HCV | - | 3.7 | 2 | 6 | Drainage | E. faecalis | E. faecalis | Recovered |
| 6 | C | LA | 75 | M | HCC | NBNC | + | 8.3 | 3 | 30 | Drainage | E. faecalis E. cloacae | No growth | Recovered |
| 7 | C | LA | 72 | M | HCC | HBV +HCV | + | 2.9 | 4 | 69 | Drainage | E. faecium | CNS | Recovered |
| 8 | C | LA | 69 | M | HCC | HCV | - | 1.5 | 3 | 385 | Drainage | No growth | No growth | Recovered |
| 9 | C | LA | 79 | F | HCC | AIH | - | 3.2 | 1 | 330 | Antibiotics | Not taken | No growth | Recovered |
| 10 | C | CC | 76 | M | HCC | Alcohol | - | 2.2 | 2 | 2 | Drainage | E. faecalis B. fragilis | E. faecalis | Recovered |
| 11 | C | LA | 64 | M | HCC | HCV | + | 1.0 | 1 | 12 | Drainage | E. faecalis E. cloacae | E. faecalis | Fatal |
| 12 | C | LA | 41 | F | MLT | NBNC | - | 3.6 | 4 | 29 | Drainage | No growth | No growth | Recovered |
| 13 | C | LA | 70 | M | HCC | HCV | - | 1.5 | 2 | 139 | Drainage | K. pneumoniae | No growth | Recovered |
| 14 | C | LA | 73 | M | HCC | Alcohol | - | 3.3 | 1 | 56 | Drainage | E. coli B. fragilis | E. coli | Fatal |
| 15 | C | LA | 62 | M | HCC | HBV | - | 0.9 | 1 | 1 | Drainage | E. faecalis P. aeruginosa | No growth | Recovered |
| 16 | S | CC | 67 | M | HCC | Alcohol | + | 2.7 | 2 | 7 | Drainage | S. pneumoniae | No growth | Recovered |
| 17 | S | LA | 66 | F | HCC | HCV | + | 1.9 | 2 | 113 | Drainage | E. aerogenes E. faecium | E. aerogenes | Recovered |
| 18 | S | LA | 64 | M | MLT | NBNC | - | 1.9 | 2 | 14 | Drainage | E. coli | No growth | Recovered |
| 19 | S | LA | 69 | M | HCC | HCV | - | 1.8 | 2 | 14 | Drainage | Not taken | K. pneumoniae | Recovered |
| 20 | S | LA | 73 | F | HCC | NBNC | - | 1.5 | 2 | 68 | Drainage | E. coli | E. coli | Recovered |
| 21 | S | LA | 74 | M | MLT | NBNC | - | 3.0 | 3 | 1 | Drainage | E. coli E. faecalis | No growth | Recovered |
| 22 | S | LA | 75 | F | MLT | NBNC | - | 4.0 | 2 | 28 | Antibiotics | Not taken | No growth | Recovered |
| 23 | S | LA | 71 | M | HCC | HCV | + | 2.6 | 3 | 3 | Drainage | No growth | E. coli | Fatal |

(*Continued*)

**Table 2.** (Continued)

| Patient No. | Group | Complication | Age | Sex | Tumor type | Etiology | DM | Maximal tumor diameter (cm) | Number of nodules (n) | Interval between RFA and diagnosis of complication (days) | Treatment for complication | Bile culture | Blood culture | Outcome |
|---|---|---|---|---|---|---|---|---|---|---|---|---|---|---|
| 24 | S | LA | 76 | F | HCC | NBNC | + | 2.6 | 1 | 571 | Drainage | S. marcescens | S. marcescens | Recovered |

Patients are listed according to the date of diagnosis. Abbreviations: AIH, autoimmune hepatitis; B. fragilis, Bacteroides fragilis; C, continuous; CC, cholecystitis; CNS, coagulase-negative Staphylococcus; DM, diabetes mellitus; E. aerogenes, Enterobacter aerogenes; E. cloacae, Enterobacter cloacae; E. faecalis, Enterococcus faecalis; E. faecium, Enterococcus faecium; HBV, hepatitis B virus; HCC, hepatocellular carcinoma; HCV, hepatitis C virus; K. pneumoniae, Klebsiella pneumoniae; LA, liver abscess; MLT, metastatic liver tumor; NBNC, non-hepatitis B, non-hepatitis C virus; P. aeruginosa Pseudomonas aeruginosa; S, single; S. marcescens, Serratia marcescens; S. pneumonia, Streptococcus pneumonia.

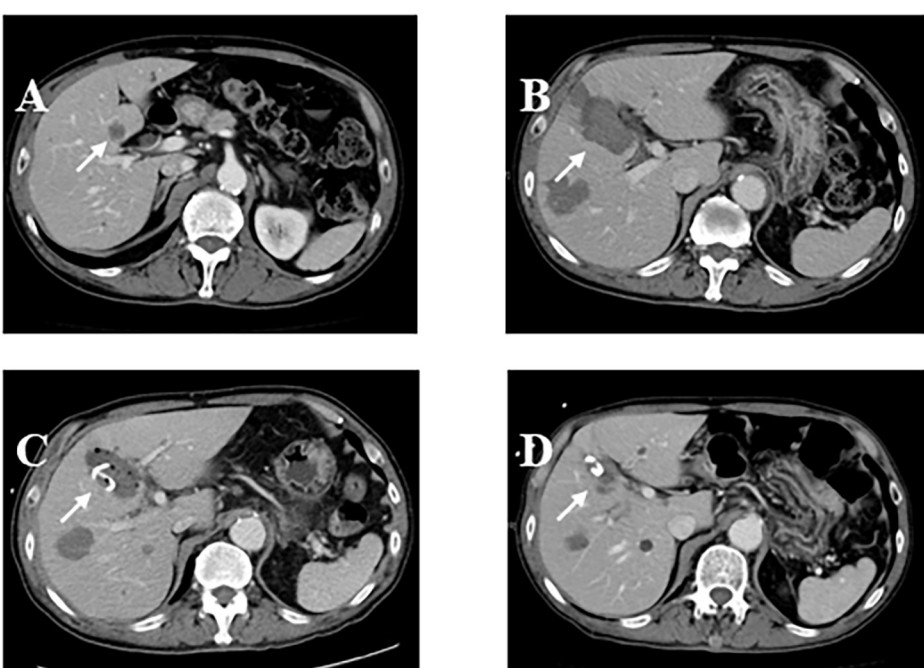

**Fig 3. A case with liver abscess following radiofrequency ablation (RFA).** A 74-year-old male patient underwent RFA of a metastatic liver tumor secondary to bladder cancer. A: Computed tomography (CT) imaging before RFA showed a 1.2 cm tumor in segment 5 (arrow). B: CT scan after RFA showed complete destruction of the tumor. C: 13 days later, the patient complained of abdominal pain with a fever of 38.5°C. CT scan revealed liver abscess on the ablated site; percutaneous transhepatic abscess drainage was performed. D: CT scan 50 days after percutaneous drainage showed shrinkage of the abscess cavity.

**Table 3. Univariable and multivariable analyses of infectious complications.**

| Variable | Univariable | | Multivariable | |
|---|---|---|---|---|
| | **OR (95% CI)** | **P-value** | **OR (95% CI)** | **P-value** |
| Age per year | 1.00 (0.96–1.04) | 0.94 | | |
| Age > 63 years | 1.99 (0.59–6.66) | 0.27 | | |
| Male gender | 0.98 (0.13–1.83) | 0.97 | | |
| HCC vs non-HCC | 0.44 (0.15–1.29) | 0.14 | | |
| Prior TACE | 2.50 (1.03–6.03) | 0.04 | 2.04 (0.81–5.09) | 0.13 |
| Maximal tumor diameter (mm) | 1.04 (1.03–1.06) | <0.001 | 1.05 (1.03–1.07) | <0.001 |
| Number of nodules | 1.11 (0.92–1.33) | 0.26 | | |
| Serum albumin | 0.65 (0.30–1.41) | 0.28 | | |
| Total bilirubin > 0.9 mg/dL | 1.21 (0.54–2.70) | 0.65 | | |
| AST > 45 U/L | 0.62 (0.27–1.42) | 0.26 | | |
| ALT > 35 U/L | 0.87 (0.39–1.95) | 0.74 | | |
| Platelet count > $11.2 \times 10^4$/mm$^3$ | 2.07 (0.88–4.83) | 0.09 | | |
| DM | 0.52 (0.23–1.20) | 0.13 | | |
| FIB-4 Index | 0.75 (0.50–1.13) | 0.17 | | |
| APRI score | 0.80 (0.52–1.25) | 0.33 | | |
| Single vs. continuous administration of antibiotics | 1.20 (0.53–2.75) | 0.66 | 2.00 (0.82–4.90) | 0.13 |

Abbreviations: ALT, alanine transaminase; APRI, aspartate aminotransferase to platelet ratio index; AST, aspartate aminotransferase; CI, confidence interval; DM, diabetes mellitus; FIB-4, Fibrosis-4; HCC, hepatocellular carcinoma; OR, odds ratio; TACE, transcatheter arterial chemoembolization; U/L, unit/liter.

**Table 4. An ad-hoc analysis with a dataset excluding overlap cases.** Univariable and multivariable analyses of infectious complications.

| Variable | Univariable | | Multivariable | |
|---|---|---|---|---|
| | OR (95% CI) | P-value | OR (95% CI) | P-value |
| Age per year | 1.03 (0.98–1.09) | 0.27 | | |
| Age > 63 years | 2.67 (0.52–9.87) | 0.28 | | |
| Male gender | 0.74 (0.26–2.08) | 0.57 | | |
| HCC vs non-HCC | 0.66 (0.15–2.89) | 0.58 | | |
| Prior TACE | 3.16 (1.18–8.44) | 0.02 | 2.11 (0.75–5.88) | 0.16 |
| Maximal tumor diameter (mm) | 1.05 (1.03–1.07) | <0.001 | 1.05 (1.03–1.07) | <0.001 |
| Number of nodules | 1.13 (0.92–1.39) | 0.24 | | |
| Serum albumin | 0.66 (0.27–1.60) | 0.36 | | |
| Total bilirubin > 0.9 mg/dL | 1.41 (0.56–3.54) | 0.47 | | |
| AST > 45 U/L | 0.65 (0.25–1.67) | 0.37 | | |
| ALT > 35 U/L | 1.02 (0.41–2.58) | 0.96 | | |
| Platelet count > $11.2 \times 10^4$/mm$^3$ | 2.73 (0.97–7.68) | 0.06 | | |
| DM | 1.97 (0.76–5.09) | 0.16 | | |
| FIB-4 Index | 0.93 (0.81–1.07) | 0.32 | | |
| APRI score | 0.85 (0.56–1.31) | 0.47 | | |
| Single vs. continuous administration of antibiotics | 0.73 (0.26–2.05) | 0.55 | 1.29 (0.43–3.88) | 0.66 |

Abbreviations: ALT, alanine transaminase; APRI, aspartate aminotransferase to platelet ratio index; AST, aspartate aminotransferase; CI, confidence interval; DM, diabetes mellitus; FIB-4, Fibrosis-4; HCC, hepatocellular carcinoma; OR, odds ratio; TACE, transcatheter arterial chemoembolization; U/L, unit/liter.

infectious complications may be lower than in intermediate- or low-volume hospitals. However, this single-center study enabled a deeper analysis of each patient who encountered infectious complications, including the bacterial species involved and the details of their clinical course.

In conclusion, the rate of infectious complications related to RFA was acceptably low. Single administration of prophylactic antibiotics did not significantly increase the rate of infectious complications related to RFA, compared with a more intensive antibiotic protocol.

## Supporting information

**S1 File. An exploratory clinical trial of prophylactic antimicrobial administration once before treatment in percutaneous radiofrequency ablation of hepatic malignancies.** (DOCX)

**S1 Checklist. TREND statement checklist.** (PDF)

## Author Contributions

**Conceptualization:** Ryosuke Tateishi.

**Investigation:** Ryosuke Tateishi, Shintaro Mikami, Taijiro Wake, Mizuki Nishibatake Kinoshita, Takuma Nakatsuka, Tatsuya Minami, Masaya Sato, Koji Uchino, Kenichiro Enooku, Hayato Nakagawa, Yoshinari Asaoka, Shuichiro Shiina.

**Methodology:** Ryosuke Tateishi.

**Supervision:** Ryosuke Tateishi, Kazuhiko Koike.

**Writing – original draft:** Ryo Nakagomi.

**Writing – review & editing:** Ryosuke Tateishi.

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
