## [Decision Letter · Decision Letter 0]

7 Apr 2021

PONE-D-21-03580

Infectious complications related to radiofrequency ablation of liver tumors: the role of antibiotics

PLOS ONE

Dear Dr. Tateishi,

Thank you for submitting your manuscript to PLOS ONE. After careful consideration, we feel that it has merit but does not fully meet PLOS ONE’s publication criteria as it currently stands. Therefore, we invite you to submit a revised version of the manuscript that addresses the points raised during the review process.

We look forward to receiving your revised manuscript.

Kind regards,

Jee-Fu Huang, M.D., Ph.D.

Academic Editor

PLOS ONE

Journal Requirements:

Reviewers' comments:

Reviewer's Responses to Questions

5. Review Comments to the Author

Reviewer #1: Overall , this article is well-written and shared good experience regarding RFA related infection complications. However, several points provided below may need to be considered before it can be accepted for publication.

1. Between February 1, 1999 and November 30, 2010, a total of 6,763 RFA treatments were performed in

2,355 patients with liver malignancy and was categorized in the group of administered continuous

prophylactic antibiotics. Since this article analyzed their data using RFA treatment times. It means that this

historical control group will have higher recount rates ( mean RFA treatment times will approach 3 for each

patient) compared to the other groups ( mean RFA treatment times less than 2 for each patient).

Higher TACE experience and tumor size was noted in historical control group, Considering the higher

recount rates in these patients when doing statistical analysis is needed to avoid mis-interpretation

especially when trying to find any risk factors which contributed to later infection complication after RFA.

2. There seemed to be a typo in page 7 . study protocol: line 4 , antibiotics until fever "redTuced " following

RFA. Please correct it.

3. It seemed not very appropriate to include infection complication occurred 300 more days later after RFA

treatment.To address antibiotic prophylaxis issue, included infection events within 30 days after RFA may

be more appropriate and realistic.

4. Previous biliary intervention history or if co-existed cholelithiasis; grade of CTP score or MELD score are not

included in the analysis , these factors could also be confounding or contributing risk factors when doing

comparison.

5. The data can't tell if antibiotic prophylaxis is really needed for HCC RFA. Authors may still should address

this issue regarding the need of antibiotics prophylaxis for HCC RFA from literature review in discussion

section.

Reviewer #2: The authors conducted a prospective study comparing the efficacy of prophylactic single-dose antibiotics with a historical control of multiple-doses antibiotics (continuous administration group) for radiofrequency ablation (RFA) of malignant liver tumors, they found that the rate of infectious complications related to RFA was comparably low in both groups. The univariate analysis has shown that prior TACE and maximal tumor diameter were risk factors for infectious complications following RFA, and only maximal tumor diameter was the only significant factor by multivariate analysis. This was a long-term observation study with a huge number of treatments, and the manuscript was well-written with adequate language. Only a few revisions were recommended in its current version.

1. This study included metastatic liver tumors hence we are wondering how many patients were primary GI-tract malignancies that might predispose to post-RFA infections because 4 in 844 developed infection episodes compared to 20 out of 9,792 in the hepatocellular carcinoma (HCC) patients?

2. Some patients received TACE before RFA, please address their baseline tumor staging in the context.

3. Since the number of tumor nodules was not a risk factor for an infectious complication after RFA by univariate analysis, I recommend the authors clarifying that why ‘the number of ablation sessions’ and ‘the duration of ablation’ were not the risk factors for post-RFA infectious complications.

4. Will the liver fibrosis stage influence the infectious complications after RFA (e.g. using FIB-4, Child-Pugh, APRI, or MELD scores instead of albumin, T. bil., AST, ALT, and platelet count)?

5. Since the post-RFA infectious complications were acceptably low under both single or multiple prophylactic antibiotics and the maximal tumor diameter was the only risk factor, will the authors recommend prophylactic antibiotics applied only to the selected patients?

6. Typo at line 4, 1st paragraph, PATIENT AND METHODS, Study protocol, ‘redTuced’, please correct it.

---

## [Author Response · Author response to Decision Letter 0]

22 May 2021

Dear Dr. Jee-Fu Huang

Thank you very much for reviewing our manuscript. We are herewith sending you the revised manuscript. We are deeply appreciative of numerous helpful suggestions by the associate editor and reviewers. We believe that the paper is much improved with revision, and we hope that you will find it acceptable for publication.

---

## [Decision Letter · Decision Letter 1]

2 Sep 2021

PONE-D-21-03580R1

Infectious complications related to radiofrequency ablation of liver tumors: the role of antibiotics

PLOS ONE

Dear Dr. Tateishi,

Thank you for submitting your manuscript to PLOS ONE. After careful consideration, we feel that it has been much improved for the potentiality of acceptance. Therefore, we invite you to submit a revised version of the manuscript that addresses the points raised during the review process.

Please respond to the statistical points raised.

We look forward to receiving your revised manuscript.

Kind regards,

Jee-Fu Huang, M.D., Ph.D.

Academic Editor

PLOS ONE

Journal Requirements:

Reviewers' comments:

Reviewer's Responses to Questions

6. Review Comments to the Author

Reviewer #1: The author had answered all the reviewer's opinion. Although the role of prophylactic use of antibiotics before RFA can't be established. This large single center retrospective analysis do provide guide for management of infection after radio frequency ablation for liver tumors.

Reviewer #2: The authors responded and revised the manuscript point-by-point to my previous comments, I have no further comments.

Reviewer #3: 1. If one patient was included in both groups for analysis, a hierarchical modeling technique may be used to account for within-subject correlation. Further, details should be added (e.g . n and % of patients included in both groups. A sensitivity analysis may be considered for a clean dataset with the exclusion of double counted patients.

2. Perhaps it is fair to add a comparison between N=348 vs N=766 during the overlapping time period.

3. Add methods used for comparisons in Table 1 to Statistical Analysis

4. Add upper limit to all reported infection complication rates

5. Use “univariable and multivariable” instead of “univariate and multivariate” throughout the text.

---

## [Author Response · Author response to Decision Letter 1]

21 Oct 2021

Dear Dr. Jee-Fu Huang

Thank you very much for reviewing our manuscript. We are herewith sending you the revised manuscript. We are deeply appreciative of numerous helpful suggestions by the associate editor and reviewers. We believe that the paper is much improved with revision, and we hope that you will find it acceptable for publication.

---

## [Editor Report · Decision Letter 2]

25 Oct 2021

Infectious complications related to radiofrequency ablation of liver tumors: the role of antibiotics

PONE-D-21-03580R2

Dear Dr. Tateishi,

We’re pleased to inform you that your manuscript has been judged scientifically suitable for publication and will be formally accepted for publication once it meets all outstanding technical requirements. I apologize for the processing delay.

Kind regards,

Jee-Fu Huang, M.D., Ph.D.

Academic Editor

PLOS ONE
---

## [Editor Report · Acceptance letter]

5 Nov 2021

PONE-D-21-03580R2 

Infectious complications related to radiofrequency ablation of liver tumors: the role of antibiotics 

Dear Dr. Tateishi:

I'm pleased to inform you that your manuscript has been deemed suitable for publication in PLOS ONE. Congratulations! Your manuscript is now with our production department. 

Kind regards, 

on behalf of

Dr. Jee-Fu Huang 

Academic Editor

PLOS ONE